# Assisted Reproduction Techniques to Improve Reproduction in a Non-Model Species: The Case of the Arabian Bustard (*Ardeotis arabs*) Conservation Breeding Program

**DOI:** 10.3390/ani12070851

**Published:** 2022-03-28

**Authors:** Janaina Torres Carreira, Loïc Lesobre, Sylvain Boullenger, Toni Chalah, Frédéric Lacroix, Yves Hingrat

**Affiliations:** Reneco International Wildlife Consultants LLC, Abu Dhabi P.O. Box 61741, United Arab Emirates; llesobre@reneco.org (L.L.); sboullenger@reneco.org (S.B.); tchalah@falconry-exps.com (T.C.); flacroix@reneco.org (F.L.); yhingrat@reneco.org (Y.H.)

**Keywords:** bustards, wild bird reproduction, artificial insemination, semen cryopreservation, conservation breeding, ex-situ conservation, sperm bank

## Abstract

**Simple Summary:**

Artificial reproduction technologies such as artificial insemination and semen cryopreservation are important tools for species conservation and long-term genetic management. As with many bird species, Arabian bustard (*Ardeotis arabs*) populations are declining and are already extinct in some regions. The International Fund for Houbara Conservation (IFHC) has started a conservation breeding program for the Arabian bustard in the United Arab Emirates (UAE). Birds were housed by pairs to allow natural reproduction. Out of 1253 eggs laid, 1090 were incubated, of which 379 were fertile (i.e., 34.8%). In total, this led to the production of 251 chicks. Due to lower than desired fertility, we introduced assisted reproduction techniques to increase fertility and develop zootechnical knowledge for the species. This paper presents the results of semen collection, artificial insemination, and semen cryobanking in Arabian bustards. Inseminations with both fresh and preserved semen led to a significant increase in fertility to 84.3% of incubated eggs (i.e., 43 out of 51 incubated eggs laid by previously artificially inseminated females). Furthermore, we confirmed the viability of cryopreserved semen and its fertilizing capacity. Our results demonstrate the usefulness of artificial reproduction techniques for the conservation of Arabian bustards and suggest that these techniques can be applied to closely related critically endangered species with a minimum of adaptation.

**Abstract:**

Artificial reproductive technologies are highly valuable for ex situ conservation. While Arabian bustard populations are declining and extinct in some parts of the range, the International Fund for Houbara Conservation in the United Arab Emirates implemented a conservation breeding program. Since 2012, a total of 1253 eggs were laid through natural reproduction, 1090 were incubated and 379 of these were fertile (fertility rate of 34.8%), leading to the production of 251 chicks. To improve fertility and acquire crucial knowledge for other endangered large birds, artificial reproduction was implemented in 2018 using fresh, refrigerated, and frozen sperm. A total of 720 ejaculates were collected from 12 birds. We analysed these samples for concentration, volume, motility score (0 to 5), viability (eosin/nigrosine), length, and morphology. The first age at collection was 35.7 ± 18.8 months, mean volume was 89.2 ± 65.3 µL, mean concentration was 928 ± 731 sptz/mL and mean motility score was 2.61 ± 0.95. Morphology analyses revealed a bimodal distribution of sperm length. Five hundred and thirty-five ejaculates were cryopreserved and the initial motility score was 3.4 ± 0.7 and 2.0 ± 0.6 after thawing, while the percentage of normal and intact membrane sperm cells decreased from 88.8 ± 7.5% to 52.9 ± 1%. Sixty-five artificial inseminations were performed, leading to a global fertility rate of 84.3%—more precisely, 85.2% and 83.3%, respectively, for fresh and cryopreserved semen. All methods successfully produced fertile eggs, indicating that artificial insemination is an efficient tool for the conservation and genetic management of the species.

## 1. Introduction

Artificial reproductive technologies provide important opportunities for assisting conservation efforts to counter the current biodiversity crisis [1,2]. Artificial insemination has become an important component of many recovery and conservation programs for non-domestic birds [3,4]. Furthermore, genomic resource banking, specifically cryobanking of gametes, is a strong component of the long-term maintenance and management of genetic diversity. It also allows individuals to contribute to the gene pool long after their death [5,6,7,8]. Hence, semen cryopreservation combined with artificial insemination provides a valuable tool for the storage and later use of sperm, increasing the capacity for robust management of genetic diversity in avian conservation breeding programs [9,10,11].

The bustard family (Otididae) is composed of 26 species, eight of which are classified as Threatened within the IUNC Red List of Threatened Species [12]. Within the family, the genus *Ardeotis* is composed of four species: the Australian bustard (*Ardeotis australis*, Gray, 1829; the only bustard in Oceania); the Kori bustard (*Ardeotis kori*, Burchell, 1822 (the second-heaviest flying bird); the great Indian bustard (*Ardeotis nigriceps*, Vigors, 1831 (one of the most endangered bird species with only 100–150 individuals remaining in the wild and listed as Critically Endangered [13,14]), and the Arabian bustard (*Ardeotis arabs*, Linnaeus, 1758). This final species is listed as Near Threatened by IUCN with a global trend of decreasing population owing primarily to hunting pressure and habitat degradation [15,16]. Four subspecies of Arabian bustards are recognized [17] (Figure 1): *A. a. lynesi* (Bannerman, 1930), which is probably extinct, *A. a. stieberi* (Neumann, 1907), *A. a. butleri* (Bannerman, 1930) and *A. a. arabs* (Linnaeus, 1758). The latter is considered extinct in Saudi Arabia, where it was included in a high-priority conservation list, ranking ninth among 102 species [18].

The Arabian bustard Conservation Breeding Program implemented by the International Fund for Houbara Conservation (IFHC) at the National Avian Research Center (NARC, Sweihan, UAE) was initiated in July 2007 with the arrival of three confiscated birds of Yemeni origin. In 2009–2010, egg collections were conducted in the Tihama region of Yemen, resulting in 11 chicks from nine harvested nests. The founding flock was thus composed of nine males and five females that were housed in groups. The first egg was laid in captivity in 2012 and the first chick produced through natural reproduction hatched in 2013 (Figure A1 and Figure A2). Between 2012 and 2021, a total of 1310 eggs were laid, with 1253 from females included in the natural reproduction program, of which 1090 were incubated. Three hundred and seventy-nine of these were identified as fertile (i.e., 34.8% fertility through natural breeding) leading to the production of 251 chicks (Figure A1 and Figure A2). In 2018, artificial reproduction techniques were implemented to improve reproductive success (number of fertile eggs and live chicks) and to investigate how best to apply these techniques in the genus. This was done while building on the experience acquired in semen collection, artificial insemination, and semen cryobanking within IFHC houbara (*Chlamydotis* spp.) conservation breeding programs [4,20,21,22], where artificial insemination yields fertility levels of 91.7% for incubated eggs.

These techniques are efficient for analyzing and sorting many semen samples and adapted to situations where both cost-efficiency and skill transfer and capacity building are a concern. The objectives of the present work on the Arabian bustard were multiple, with the first to increase the proportion of fertile eggs within the conservation breeding program. Indeed, natural reproduction would not be sufficient to ensure both reproduction in captivity and an efficient long-term genetic management of the species in captivity through pair selection. In addition, this non-model species can be a surrogate to develop, test, and provide techniques that can be applied to other large, endangered birds such as the Great Indian bustard, therefore efficiently contributing to their conservation.

Thus, the present paper aims to present the first data on routine analyses of semen characteristics within the *Ardeotis* genus as well as fertility outcomes resulting from the use of assisted reproduction techniques, including a cryobank of cryopreserved semen. More specifically, we provide information on female laying patterns, semen collection, qualitative and quantitative evaluation of semen, sperm morphology, artificial insemination and subsequent fertilization success and hatching rates.

## 2. Materials and Methods

### 2.1. Experimental Birds and Breeding Program Management

Adult housing facilities consisted of elliptical tunnels measuring 9 × 15 m (Figure A3) where food and water were provided ad libitum. Pairings were constituted while considering optimal maintenance of genetic diversity within the captive flock using pedigree analyses [23,24,25]. The natural pairing was not considered, as it would have required prohibitively large pens and would not allow strict pairing management to ensure long-term maintenance of genetic diversity. Males and females were kept together throughout the year and eggs were checked and recorded daily. The breeding flock in January 2022 is composed of 147 individuals (72 males and 75 females). While eggs were recorded throughout the year, most were laid between March and August. Thirty-eight females laid an average of 13.8 eggs per year (range 1 to 44 eggs per layer and per year) for a total amount of 1310 eggs. The average clutch size was 1.55 ± 0.67. Eggs from a single clutch were laid at 2 days intervals (median = 2 days, average = 2.3 ± 0.7 days), while the inter-clutch interval was 12 days (median = 12 days, average = 16.5 ± 16.7 days). 

Upon transfer to incubation, all eggs were disinfected and controlled for defects before being placed in incubators. Water loss was calculated by measuring the change in egg weight throughout incubation. Fertility and viability were measured via egg candling on days 3, 9, 12 and 20 of incubation. Systematic necropsy of non-hatched eggs allowed the team to analyze if the eggs were fertilized or not. Only incubated eggs were considered when evaluating egg fertility results. The hatching rate was evaluated as the proportion of viable live chicks hatched over the total number of incubated eggs. The average incubation duration was 21.8 ± 0.7 days. The rearing of future breeders was developed to minimize stress and promote familiarity with humans; this ensures a limited amount of stress during captures such as artificial insemination or vaccinations. Upon hatching, chicks were transferred to rearing facilities where they were imprinted before being transferred to adult enclosures at about one year old. Imprinting consisted of frequent interactions between chicks and keepers through visual or acoustic contact, hand-feeding, or physical interactions, such as being held. Imprinting reinforcement occurs throughout an individual’s life to reduce stress and allow the implementation Of artificial reproduction techniques.

The Abu Dhabi authorities approved the breeding program and captivity conditions under reference 116620140. Founders were transferred to the facilities from Yemen under CITES certificate number 10MEW3077. Operations are conducted in a humane manner, and each breeding protocol is developed while ensuring the appropriate and ethical treatment of birds. To minimize stress during capture, birds are fitted with hoods during manipulation. Furthermore, all operations requiring handling of the birds were performed only by the most experienced keepers under the supervision of veterinarians. The facilities are designed to ensure the well-being of the birds and prevent injuries. On-site veterinary facilities provide the best possible care for sick or injured birds by a team of expert veterinarians, and the standards from sanitary authorities are regularly controlled.

### 2.2. Artificial Reproduction

#### 2.2.1. Semen Collection

Semen collections were performed yearly from 2018 to 2021, between January to September (Figure A7). Semen was collected every alternate day from displaying imprinted males using the technique described in Saint Jalme et al. [4]. Briefly, a dummy female was presented to the male. Once the male mounted the dummy and started copulating, a glass dish was used to collect the ejaculate by pressing gently on the male’s cloaca during ejaculation (Figure A5). This technique recapitulates semen collection and characteristics more closely to in vivo conditions while addressing the welfare concerns of the birds [26]. The semen was immediately transferred to a 2 mL cryotube, maintained at room temperature (RT~24 °C), and transferred to a nearby laboratory to be analyzed and prepared for artificial insemination or preservation. 

#### 2.2.2. Semen Analyses

Semen analyses methods were based on routine procedures developed for houbara bustard conservation programs where the average number of ejaculates analyzed per day reached 593 with a maximum of 1303 samples in 2021. In such cases, simple but reliable and robust methods were implemented to ensure all ejaculates were analyzed but also to permit an easy transfer of skills and allow for capacity-building. 

Arabian bustard samples were processed in this same laboratory by experienced technicians. Upon arrival, fresh ejaculates were immediately analyzed, all procedures were done at room temperature, and all used materials were kept in the same conditions (RT~24 °C). The mass motility index was assessed by placing a 5 µL droplet on a glass slide (10× phase contrast objective—Nikon E200 phase-contrast equipped microscope) and evaluated using a mass motility index [4]. This visually evaluated index has been shown to be well-correlated with fertility [27,28,29]. Sperm motility was scored from 0 to 5 in the periphery of the droplet, using the following scale: (0) total lack of movement, (1) mostly non-motile sperm with the few motile sperm lacking forward movement, (2) less than 50% of sperm showing moderate activity, (3) above 50% motile sperm showing forward movement, (4) almost all sperm showing fast forward movement, and (5) almost all sperm showing fast forward movement with waves and whirlwinds [4]. Regular controls were performed to ensure the motility scoring was standard throughout the project.

The same slide preparation was used to evaluate the occurrence of contamination within ejaculates (e.g., feces, urates, blood cells, etc.); contaminated ejaculates were discarded and were neither analyzed nor considered to be used for artificial inseminations. Among the remaining non-contaminated samples, the volume of undiluted ejaculates was measured (±1 μL) using an electronic pipette. After dilution with Lake 7.1 diluent (dilution 1:1) [30], the concentration of each ejaculate was assessed by light absorption of semen with a photometer (Jeanway 6051 colorimeter, Jeanway Ltd., Dunmon, UK) at a wavelength of 600 nm, and results were given in millions of spermatozoa per milliliter (10^6^/mL) [31]. The number of sperm per ejaculate was calculated based on volume and concentration.

The proportion of morphologically normal sperm (intact acrosome, normal head shape, non-swollen heads, normal midpiece and tail, non-teratologic cells) and intact membranes was simultaneously accessed by the eosin-nigrosin (EN) method where intact cells appear white while membrane-damaged cells will incorporate the eosin and show pink under staining [32].

Sperm length was accessed placing a 5 µL droplet of diluted and fixed semen sample (Hancock solution [33]) between the slide and a 22 × 22 mm coverslip (wet mount). Sperm cells were photographed and the length of the acrosome, head, midpiece and tail measured only on morphologically normal sperm using cellSens image software (version 1.12, Olympus\CellSens Software, Tokyo, Japan).

#### 2.2.3. Semen Refrigeration for 24 h

While most unused ejaculates were cryopreserved (cf. below), a few ejaculates were refrigerated for 24 h to be used for artificial insemination the next day to follow the female laying pattern closely and ensure optimal fertility potential (*n* = 6). After qualitative and quantitative analyses, including dilution with Lake 7.1 diluent (dilution 1:1), these ejaculates were preserved 24 h at 4 °C in 2 mL Eppendorf Safe-Lock Tubes when the preserved volume was below 250 µL, while 6 mL Sterlin scintillation vials were used otherwise. Motility scores were recorded before and after refrigeration and ejaculates were placed on an orbital shaker plate with gentle shaking to allow the ejaculate to reach room temperature prior to analysis or artificial insemination [22]. Motility results were compared by a Wilcoxon paired rank-test (*p ≤* 0.05).

#### 2.2.4. Semen Cryopreservation and Thawing

Most of the collected and non-contaminated ejaculates were submitted to the freezing protocol. Ejaculates were frozen based on Tselutin’s procedure [34]. This procedure was selected for its simplicity and known efficiency with bustard semen. Additionally, that procedure does not require the cryoprotectant to be removed by centrifugation before artificial insemination [34]. Ejaculates were cooled at 4 °C for 30 min, supplemented with dimethyl-acetamide (DMA) to a final concentration of 6%, gently mixed and kept for 1 min in an iced water bath. Then, the ejaculate was quickly dropped into liquid nitrogen to form frozen pellets. These pellets were transferred into cryovials and cryopreserved in liquid nitrogen.

During the thawing procedure, pellets were quickly placed on a thermo-regulated stainless conical hotplate (Figure A4) set at 60 °C [34]. The equipment allowed for the immediate evacuation of the thawed pellets into a 5 mL glass beaker. Motility viability/morphology analyses were repeated after thawing. Motility results were compared by a Wilcoxon paired rank-test and viability/morphology with a paired *t*-test (*p* ≤ 0.05).

#### 2.2.5. Artificial Insemination

Based on the experience acquired with houbara bustards, female laying history was taken into consideration to decide the most appropriate time to catch and inseminate the females to optimize fertility while minimizing the number of artificial inseminations performed to reduce stress. Laying histories were carefully recorded, so the artificial inseminations were generally performed within 3 to 6 days before the next clutch [4]. Specifically, designed specula were used to open the cloaca while sperm was slowly delivered at the beginning of the oviduct using a 250 µL positive displacement insemination micropipette [4] (Figure A6). Artificial inseminations were performed with a minimum of 10 million spermatozoa for a maximum volume of 250 µL.

## 3. Results

### 3.1. Semen Collection and Analyses

Between 2019 and 2021, semen collection was attempted for 34 males, of which 13 were collected for a total of 720 ejaculates. All collected males were captive-bred individuals that were imprinted. The number of ejaculates collected per bird, age in months at first collection, the ejaculate volume, concentration, sperm number and motility score are summarized in Table 1. Details regarding individuals and yearly semen collection results and monthly variations of semen characteristics are presented in Appendix A (Figure A8 and Table A1). 

The mean of the total sperm length was 59.74 ± 3.02 µm (52.61 to 68.38 µm (Figure 2)). Each section measurements were acrosome 2.22 ± 0.22 µm (1.54 to 3.97 µm), head 8.79 ± 1.38 µm (6.76 to 15.04 µm), midpiece 3.35 ± 1.12 µm (2.17 to 8.33 µm) and tail 45.37 ± 2.05 µm (40.37 to 49.93 µm). Proportionally, most variation occurs in head sizes where a bimodal pattern of sperm sizes is the result. Some ejaculates consist of sperm with heads that were twice as long as others (Figure 2). A boxplot and violin histogram of sperm-length measurements can be found in Appendix A (Figure A9).

### 3.2. Semen Refrigeration for 24 h

Six ejaculates were refrigerated for 24 h for further use for artificial insemination. The mean preserved volume was 231.3 µL ± 75.5 and the mean motility significantly decreased from 4.08 ± 0.49 to 3.33 ± 0.52 after refrigeration (*p ≤* 0.05). No morphology records were taken after refrigeration.

### 3.3. Semen Cryopreservation

A total of 535 ejaculates were cryopreserved; these represented twelve males. Twenty-four ejaculates were thawed for artificial insemination and to evaluate the impact of the cryopreservation method on cell viability and morphology. Thirteen samples representing seven males were reanalyzed for viability/morphology after thawing. The average motility decreased significantly from 3.4 ± 0.7 to 2.1 ± 0.6 (*p* ≤ 0.001) after cryopreservation, and the proportion of live/normal sperm decreased significantly from 90.3 to 52.9% (*p* < 0.05, Table 2, Figure 3).

### 3.4. Artificial Insemination

Fifteen females were inseminated for a total of 65 artificial inseminations with an average of 3.6 ± 3.3 artificial inseminations per female and per year (range 1–10). Forty-one inseminations were performed with fresh semen, five with semen preserved for 24 h and 19 with frozen/thawed semen. These artificial inseminations produced 33 eggs; six were preceded by artificial insemination with cryopreserved semen and only five were deemed to be fertile (i.e., 83.3%). Twenty-seven eggs were preceded by artificial inseminations with fresh semen, only of which 23 were fertile (i.e., 85.2%) (Table 3).

The remaining eighteen eggs were preceded by a combination of artificial inseminations with a different type of semen; fifteen of them were fertile (i.e., 83.3%). Thus, 51 incubated eggs were preceded with different semen-type artificial inseminations of which 43 were fertile (i.e., 84.3%) at the ninth day of incubation.

## 4. Discussion

This study is the first to present data on the semen characteristics of the Arabian bustard and for the genus *Ardeotis*, a genus where each species in the genus has been the subject of a conservation breeding program [35,36,37]. Here, we present the results of a routine assessment of mass motility index, concentration, volume and some complementary evaluation of viability, morphology, and length.

Our results demonstrate the potential for artificial reproduction techniques to conserve Arabian bustards and suggest these can be applied, with a minimum of adaptation, for closely related endangered species, such as the Great Indian Bustard. Indeed, the use of artificial insemination led to a significant increase in fertility compared to natural breeding (an increase from 34.8% to 84.3%). Furthermore, implementing an affordable, easy to execute, and robust pellet cryopreservation protocol allowed the development of a cryobank where all donors from the captive population are represented, acting as a substantial tool for the long-term genetic management of the species.

However, differences occur between species requiring continuous research, development, and adaptation. For example, the initial semen collection attempts involved stroking techniques performed in cranes, partly because of the similar size and morphology of both species [3,38] (Figure A5). However, this collection method was not adequate due to the pelvis configuration of bustard males. Then, from the knowledge acquired through experience with both North African (*C. undulata*) and Asian (*C. macqueenii*) houbara within IFHC conservation breeding programs, we then decided to proceed with semen collection using a dummy female (Figure A5). This required that males had to be habituated to a human presence to reduce stress and ensure they would mount a dummy with a keeper close by. Because birds will respond to the imprint process differently, and some will fail to approach the dummy or ejaculate under these conditions, this required an adjustment of rearing protocol to ensure most males can access reproduction. 

Our findings demonstrate that improved zootechny is likely the reason for successful sperm donation from younger males during the latest cohorts (i.e., 2017 and 2018; Table A2). This also likely explains why no semen has been collected so far from wild founders. Importantly, our method is the closest to natural reproduction in that it allows males to display and approach the dummy freely without forcing ejaculation. This is beneficial in terms of animal welfare as it reduces the stress to the males [26] and because semen characteristics are likely to be closer to what can be expected in natural populations, leading to more reliable ejaculate analyses.

Little is known about the full extent of the breeding season of the species in the wild. Reproduction has been recorded from April to June in the most Western part of the range, extending possibly to October or November in Yemen [17], and between July–August in Niger (IFHC personal records). At the NARC, breeding was recorded throughout the year, with 86% of eggs being laid between March and August (Figure A7). Nevertheless, precautions must be taken when defining the species’ breeding season on that basis, as the NARC is located outside its natural range and birds are receiving optimal care (i.e., *ad libitum* food and water, veterinary care). Still, the extent of the breeding season associated with small variations in semen characteristics suggests that the species is highly opportunistic and able to take advantage of favorable conditions. A similar pattern was reported for the Great Indian bustard, with breeding recorded in all months of the year (highest between March and September), with rains apparently triggering reproduction events [39].

Our results indicate that Arabian bustard sperm is longer than Asian houbara (*Chlamydotis macqueenii*; mean length = 47.92 ± 4.69 µm, (unpublished data). It has been hypothesized that inter-species differences in avian sperm length are the result of differences in structural variations related to midpiece mitochondrial composition or in nuclear organization and chromosome elongation [40], while other hypotheses suggest that differences in head length are the result of polyploidy and extra DNA content [32]. Nevertheless, most hypotheses postulate that differences in sperm competition intensity drive variation in sperm length, with longer sperm having greater swimming velocity [41,42]. Still, a metanalysis in pheasants (Phasianidae) [43] indicated that sperm size was negatively correlated with the duration of sperm storage in the female reproductive tract, estimated as a function of clutch size. That study concluded that species with larger clutches might be expected to store larger quantities of sperm since they have more eggs to fertilize [44,45], and sperm size was not correlated with the intensity of sperm competition (as measured by testis size) [43]. Asian houbara clutch size ranges from two to four eggs depending on latitude [46,47], while clutches for Arabian bustards are one to two eggs [17]. As Otididae is a family where sexual competition is central to reproduction [48], further data could provide insight into mechanisms driving differences in sperm morphology.

In addition, we observed a bimodal distribution of sperm size in Arabian bustards’ ejaculates, with some sperm being twice as long as others; this difference is primarily due to longer sperm heads. Intra-individual variations in sperm size have been presented as a potential consequence of reduced sperm competition, therefore, relaxing selection on optimal sperm length [49,50]. Although it is typically believed that longer sperm swim faster, a study on the Canary Islands’ chiffchaff (*Phylloscopus canariensis*) found that longer sperm were slower, therefore not supporting the sperm morphology–swimming speed hypotheses presented by other groups where longer sperm were faster swimmers [51]. Thus, the relationship between sperm morphology and sperm motility should be investigated further with the Arabian bustard.

While the presented results are the only semen characteristics data reported for the genus, data are available from two related species, the North African and Asian houbara, both bred as part of conservation breeding programs managed by the IFHC. Mean motility, volume, and proportion of normal sperm were similar, but both concentration and number of sperm are considerably higher in Arabian bustards (Table 4). In bustards, species body size did not correlate with greater sperm concentration, and large intra-family concentration variations have also been observed in Phasianidae [8,43]. Sperm competition and associated testis size have been suggested as drivers of sperm concentration variation [52]. Unfortunately, relevant anatomical records are lacking for bustards, hindering the investigation of the correlation between testis size and sperm competition (for a review [42]).

A viable and competent sperm population is important for successful species reproduction, especially considering the necessity of physiological polyspermy for successful avian embryo development [54,55]. Hence, further studies are necessary to objectively evaluate sperm function and movements on fresh, refrigerated, and cryopreserved samples. Sperm velocity can be measured through computer-assisted analyses, such as CASA [56], while sperm function analyses with flow cytometry can be obtained by DNA compaction through the sperm chromatin structure assay (SCSA) [57], sperm mitochondrial function [58], membrane integrity through a fluorochrome combination of propidium iodide (PI) and SYBR 14 [59], or reactive oxygen species (ROS) [60]. These parameters will provide cues on sperm quality required to ensure optimal conservation methods and the provision of the maximum number of sperm available for fertilization [8,61].

While using accessible sperm analyses methods, our results demonstrate the efficiency of assisted reproduction techniques in improving reproduction in captive Arabian bustards. Indeed, egg fertility results increased from 34.8% of incubated eggs with natural reproduction to 84.3% with artificial inseminations. These results were comparable to those routinely obtained on a larger scale with houbara bustards where fertility reaches 91.7%. Our results also illustrate cryopreservation’s effectiveness in providing ejaculates with sufficient fertilizing power. Tselutin’s pellet method, using DMA as a cryoprotectant, proved to be a robust cryopreservation method. It has the advantage of being relatively easy and economical to implement, both in terms of freezing and thawing protocols and in terms of required equipment. It then facilitates the implementation of a semen cryobank for the species as for other avian species [8].

## 5. Conclusions

Conservation breeding programs play a key role in supporting in-situ conservation, and assisted reproduction techniques are a major enhancing factor in ensuring that demographic and genetic goals can be achieved. Here, we demonstrate the potential of these techniques for the conservation of Arabian bustards and closely related and highly endangered species. Artificial insemination induced a significant increase in the percentage of fertile eggs, including with cryopreserved ejaculates. Further studies will be required to investigate and optimize sperm quality evaluation, eggs fertility, and hatching rate results (i.e., optimal sperm collection intervals, the minimum amount of sperm used during the artificial insemination, storage duration within the female sperm storage tubules, optimal methods for sperm evaluation, egg incubation protocols, etc.).

On a broader scale, our results also indicate that valuable information for artificial reproduction of endangered bird species can be obtained with a limited laboratory setup. The use of the methods presented here could promote the wider use of assisted reproduction technologies in endangered avian species, from artificial insemination to genome resource banking.

## Figures and Tables

**Figure 1 animals-12-00851-f001:**
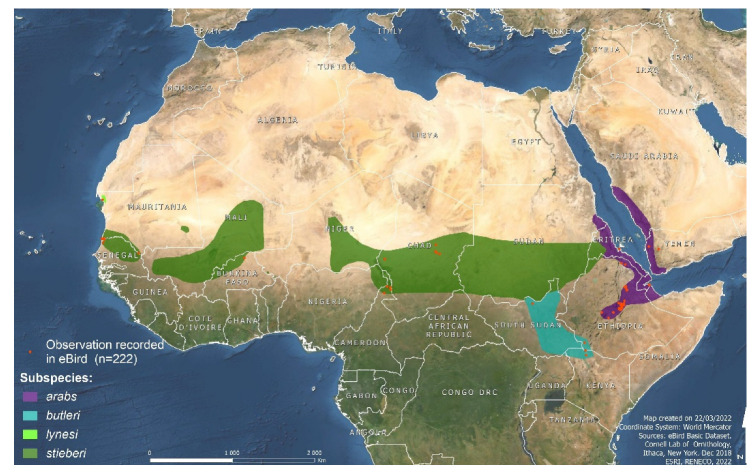
Distribution of Arabian bustards (*Ardeotis arabs*) subspecies. Red dots are observation events as recorded in eBird [19].

**Figure 2 animals-12-00851-f002:**
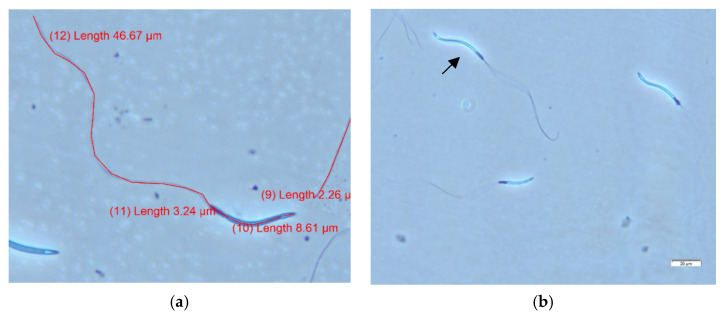
(**a**) Arabian bustard sperm measurements of the acrosome, head, midpiece and tail on cellSens image software, numbers between parentheses are generated by the software to count segments, and the length is represented in micrometers (version 1.12, Olympus cellSens Software); (**b**) Arabian bustard sperm showing a teratogenic defect (double-tail, highlighted by the arrow) and differences in sizes (Hancock solution fixation, wet chamber method, phase contrast 1000× oil, Olympus BX41).

**Figure 3 animals-12-00851-f003:**
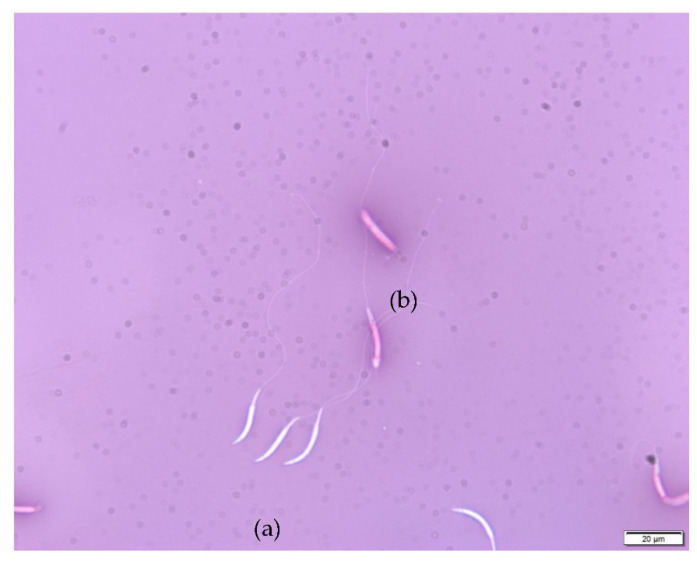
Frozen-thawed Arabian bustard EN/morphology and viability sperm evaluation: (**a**) the membrane of uncoloured sperm is considered as intact, and (**b**) damaged sperm membrane is positive for eosin coloration, therefore coloured in pink (Olympus BX61, 400× brightfield).

**Table 1 animals-12-00851-t001:** Overall information on semen collection from 2019 to 2021. Only ejaculates that were not contaminated with urates, feces, or red blood cells were considered for overall semen characteristics. Means are presented along with standard deviations.

	Mean	Range
Number of males with collection attempts	34	
Number of donors	13	
Age at first collection (in months)	36.3 ± 18.4	10–61
Number of ejaculates	720	
Number of ejaculates per donor	55.4 ± 40.6	1–129
Number of ejaculates per donor and per year	25.7 ± 26.7	1–84
Non-contaminated ejaculates	343	
Volume (µL)	87.5 ± 54.1	4–290
Motility	2.9 ± 0.9	0–4.5
Concentration (10^6^/mL)	1091.9 ± 803.6	23.9–4315.3
Sperm number (×10^6^)	83.18 ± 78.7	1.6–672.8

**Table 2 animals-12-00851-t002:** Results of sperm viability and morphology analyzes for 13 ejaculates before and after cryopreservation. Means are presented along with standard deviations, and values are presented in percentage.

		Fresh	Frozen/Thawed	*p*-Value
Normal live	Mean	90.3 ± 8.2	52.9 ± 11	
	Range (min/max)	76.5–99.9	31–71	*p* ≤ 0.001
Normal dead	Mean	3.4 ± 2.4	38.6 ± 10.2	
	Range (min/max)	0–8	23–57	*p ≤* 0.001
Abnormal live	Mean	8.3 ± 5	4.2 ± 3.5	
	Range (min/max)	3–17	0–11	*p ≤* 0.05
Abnormal dead	Mean	1.2 ± 1.5	4.2 ± 3.7	
	Range (min/max)	0–4.2	0–10	*p ≤* 0.01

**Table 3 animals-12-00851-t003:** Mean parameters of artificial inseminations performed with fresh, 24 h refrigerated and cryopreserved semen. Means are presented along with standard deviations.

	Fresh	24 h Refrigerated	Cryopreserved
Artificial Inseminations	41	5	19
Volume (µL)	209.4 ± 62.0 µL	211.0 ± 72.5	183.6 ± 72.5
Number of sperm (×10^6^)	73.6 ± 34.9	78.3 ± 21.8	66.6 ± 33.5
Motility	3.3 ± 0.7	3.4 ± 0.5	1.9 ± 0.4
Eggs	27	0	6
Fertile Eggs (9th day of incubation)	23	0	5
Fertility rate	85.2%		83.3%

**Table 4 animals-12-00851-t004:** Average semen characteristics for three bustard species bred within IFHC Conservation breeding programs: Arabian bustard (NARC, this study), North African houbara (ECWP, Morocco, unpublished data) and Asian houbara (NARC, unpublished data).

	Arabian Bustard (This Study)	North African Houbara	Asian Houbara
Ejaculates	719	852,714	397,877
Volume (µL)	89.2	82.1	63.1
Motility	2.6	2.9	2.8
Concentration	928.7	401.4	423.5
Number of sperm	72.5	31.1	26.3
Proportion of normal sperm	0.89	0.80 [53]	n/a

## Data Availability

Data sharing not applicable; all data have been shared in the article.

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
