# Peer review of "Assisted Reproduction Techniques to Improve Reproduction in a Non-Model Species: The Case of the Arabian Bustard (*Ardeotis arabs*) Conservation Breeding Program"

_animals, 2022, doi:10.3390/ani12070851_

Round 1
Reviewer 1 Report
This manuscript is important and will be of value to avian conservationists when significantly revised.
The primary issues include the failure to separate the fertilization data for fresh, refrigerated and cryopreserved sperm; the minimal discussion regarding these three treatments; and the greatly overemphasized discussion about sperm morphology, which was not a factor analyzed in the success of the artificial inseminations.

Reviewer 2 Report
This article is a part of a series of studies on the protection and preservation of endangered or extinct species. Biotechnology is fully justified in the case of unique species listed on the IUCN Red List of Threatened Species. The aim of the present study is interesting, but in my opinion methods for assessing sperm quality are not sufficient. This is the reason why the article should be rejected for publication in Animals.
Reviewer 3 Report
Information on any species that is threatened or near threatened or on the endangered list needs to be taken seriously. The authors have presented a large body of solid information with an impressive number of ejaculates analyzed. Work of this kind provides important baseline information and needs to be encouraged and should be published in the long term.
However, despite these positive comments the paper needs to be seriously revised and major changes need to be made. Here follows some minor and major comments and suggestions to improve the paper in chronological form:
line 48: use of reproductive cell makes no sense and rephrase
line 60: find correct reference source
line 79: technics is wrongly used throughout the text and should read: techniques or reproductive biology technologies
Line 80: Be careful of the usage of "fertility rate". Rather improve reproductive outcome as measured by "live chicks hatched". I really miss this throughout the text that the authors focus on the term fertile eggs etc. but it must be more succinctly defined as chicks hatched and if that was not the endpoint it needs to be defined what is meant by "fertile". In human artificial conception success is measured as: live birth of a NORMAL baby with no defects. This may sound like playing with words but it is vitally important that we clarify clearly what is meant and what the so called successful endpoints are before we talk about fertility.
Line 98: Pairing constituted - genetic grounds. Were no natural pairing considered as that would have largely excluded female cryptic choice?
Line 109 relating to fertility - hatched chicks/semen quality? The same issues as raised for line 80
Line 129: No, we simply do not know that it closely approached the volume ejaculated during natural copulation. Rather state that the dummy method constitutes semen collection and characteristics more closely to in vivo conditions
Line 130: Temperature of vial?
Line 132 dealing with semen methodology: This is really insufficient and needs considerable more detail. Temperature control, slide warming, type of slide and cover slip, volume of droplet analyzed Microscope type, phase contrast settings - all the details are required. Furthermore, while the authors later in the discussion address the importance of future studies involving CASA the current manual, very subjective analysis with no quality control and outdated technology should have been avoided. The authors need to indicate if analysis have been performed in duplicate and what quality control measures have been taken into consideration (e.g. two technicians scoring double blind independently?)
Line 150: Sperm size is "assessed" not accessed and really the wrong terminology. It is assumed that the authors refer to sperm morphometrics and measuring the LENGTH (not size as this may also be volume) of the different sperm components such as head, midpiece and tail. Why not width? More detail is required exactly how this is performed using fresh/fixed sperm and smear method and the correct details.
Line 181: Yes the authors used a 250ul pipette but the volume inseminated should also be stated here and not only in the Tables
After Line 204: Fig. 1 a) (b), (c) and (d) typically show that the values and ranges have very little meaning over the breeding period. It is only when sperm functionality and sperm sub-populations are measured using e.g. CASA technology that meaningful variations and correlations can be discovered over the breeding period. What has been depicted here are values with a very "high noise level" and nothing more than a broad range of values that does not seem to point to any peaks.
Line 223: Length not size
Lines 231 to 235; No rationale provided for reanalysis viability/morphology
Lines 251 to 256: This is really too diffuse. Be specific by putting these in Tables with required SD's etc.
Line 258: Needs to be emphasized in studies of this kind we really need to assess sperm functionality using a wide range of preferably quantitative tests rather than outdated subjective manual methods which have been shown in many studies to be inconsistent.
Line 289: November
Line 293: extent and while indicating this the paper requires serious English editing
Line 305: correlate
Line 308: There are many other characteristics such as sperm structure etc. that relates to sperm competition and not just testis size
Line 313: Length not size
Line 366 to 368. Rephrase as it makes no sense
Conclusion: The English and final comments are not good. The Conclusion must be rewritten and also keep in mind the "fertility" problem previously mentioned
Reviewer 4 Report
The manuscript “Applying artificial reproduction technologies and cryobanking for Arabian bustard (Ardeotis arabs) conservation breeding” aims to present the development of artificial insemination associated with semen conservation technology as a tool for the conservation of the Arabian bustard. Some points need to be better presented for a better understanding of the proposal of the manuscript:
- The way the title is presented does not specify what the authors sought to develop. Only at the end of the introduction does the proposal of the manuscript really become understandable. I think abstract titles should be better presented for the experimental purpose of the study.
- In the introduction, information about the importance of sperm cryopreservation is for any species. The authors need to clarify what are the hypotheses for the species of study, what are the limitations in this species.
- Authors must describe in the methodology all analyzes performed on sperm, such as morphometric analyses.
- In “the mean motility decreased from 228 4.1 ± 0.5 to 3.3 ± 0.5 after refrigeration”, Was there a statistical difference? Authors need to detail this information in the methodology and results.
Specific Comments:
- I suggest changing the term "artificial reproduction technologies" to "assisted reproduction techniques". In case the authors want to keep this terminology, I suggest referencing since the term "assisted reproduction techniques" is more commonly used.
- In Simple Summary and Abstract, to define “UAE”.
- In Simple Summary, authors must specify whether the "cryobanking" covered will be sperm only.
- In Abstract, enter the value unit “52.9±11”.
- In methodology, “Semen collections were performed between January to September.” Which year?
- Insert information about approval of the study in the institutional ethics committee.
Round 2
Reviewer 1 Report
The excellent response of the authors to the reviewers' suggestions has greatly improved this manuscript. It is an interesting and important project.
The following English language improvements will further clarify the manuscript.
L14-15 this is still confusing. ‘fertility’ must be defined. I believe the authors are saying that of 1253 eggs laid, 251 hatched. Of 1044 eggs incubated, 379 were fertile. However, this may not be correct as 1044/379 is 36.3%.
L19 this new number (84.3%) is correct, but one wonders how the 88.2% in the original manuscript was calculated.
L20 does ‘previously’ inseminated females mean those hens that had been artificially inseminated or hens that had been inseminated previously by any method (naturally or artificially)?
L15 and L28 state that 1044 and 1090 eggs, respectively, were laid. This inconsistency must be corrected.
L37-40 it would be most informative if the authors would give fertility rates for AI with each of the treatments (fresh, cryopreserved and refrigerated sperm).
L51 combined with rather than combined to
L67 and 71. These references have been reversed. Ref 18 is eBird.
L90 what is meant by “skill transfer is a concern”?
L90 clarify that the subject is now the Arabian bustard program (not the IFHC program)
L121 replace ‘autopsy’ with ‘necropsy’. Autopsy refers to human post-mortem examination, whereas necropsy refers to animal post-mortem.
L128 what is an example of a ‘physical interaction’?
L128 and 130 what is the difference between rearing and imprinting?
L136 replace ‘human’ with ‘humane’
L146 replace ‘alternative’ with ‘alternate’
L148 replace ‘starts’ with ‘started’
L162 replace ‘are immediately’ with ‘were immediately’ and ‘are done’ with ‘were done’
L167 what is the relevance of sexual selection studies in this manuscript?
L183 better to say ‘normal sperm (intact acrosome…)’
L184 what is a ‘non-swallowed head’?
L186 replace ‘appears’ with ‘appear’
L191 replace ‘morphological’ with ‘morphologically’
L197 were the ejaculates diluted with semen extender before refrigeration?
L208 what were the ejaculates diluted with?
L256 replace ‘during’ with ‘for’
L261 change to ’24 ejaculates were thawed for insemination,…”
L262 change to ‘…morphology. Thirteen samples, representing seven males, were’
Table 2 indicate somewhere in the table title or in the table itself that these values are %.
L277 end the sentence at ‘semen’. Begin the next sentence ‘These inseminations…’
L292 change to ‘..genus has been the subject…’
L304-322 this is a very important point!
L365 what is meant by ‘..subsequent number of sperm…”?
L414 change ‘…assist-ed’ to assisted
Reviewer 3 Report
I am satisfied that the authors have made the required changes
Author Response
Comments and Suggestions for Authors
I am satisfied that the authors have made the required changes
- We are grateful to the reviewer for her/his appreciation and acknowledgement of the importance of our work. We revised our manuscript according to the comments provided. We are grateful to the reviewer for her/his contribution to improve our manuscript.